# LOSI: Improving Multi-agent Reinforcement Learning via Latent Opponent Strategy Identification

## Abstract

In collaborative Multi-Agent Reinforcement Learning (MARL), agents must contend with non-stationarity introduced not only by teammates' concurrent decisions but also by partially observable and diverse opponent strategies. Although recent MARL algorithms have achieved strong performance in complex decision-making tasks, they often overfit to specific opponent behaviors, resulting in sharp performance drops when encountering previously unseen strategies. To overcome this limitation, we propose Latent Opponent Strategy Identification (LOSI), an unsupervised framework that identifies and adapts to opponent strategies in real time without requiring explicit supervision. LOSI employs a trajectory encoder trained with a contrastive learning objective (InfoNCE) to map opponent behaviors into compact and discriminative embeddings. These embeddings are then used to condition both the MARL policy and the mixing network, thereby enabling adaptive and robust decision-making. Experimental results on challenging SMAC-Hard scenarios with mixed opponent strategies demonstrate that LOSI substantially improves generalization and achieves competitive or outperforming results compared to strong MARL baselines. Further analysis of the learned embedding space reveals meaningful clustering of trajectories by opponent strategy, even in the absence of ground-truth labels.

## 1 Introduction

Multi-Agent Reinforcement Learning (MARL) has achieved remarkable success across complex domains, ranging from video games such as StarCraft II to real-world applications including autonomous driving and multi-robot coordination. To tackle the challenges of coordination and scalability, numerous MARL methods have been developed, focusing either on value decomposition (Sunehag et al., 2017; Rashid et al., 2018; 2020; Wang et al., 2020a) or on cooperative exploration (Yang et al., 2020; Mahajan et al., 2019; Wang et al., 2020b). Among them, value-based methods (Sunehag et al., 2017; Son et al., 2019) have demonstrated particularly strong performance in benchmark tasks such as the StarCraft Multi-Agent Challenge (SMAC) (Samvelyan et al., 2019).

A fundamental challenge in collaborative MARL, however, lies in the ability of agents to generalize to diverse opponent behaviors. While state-of-the-art algorithms such as QMIX (Rashid et al., 2018), MADDPG (Lowe et al., 2017), and MAPPO (Yu et al., 2022a) can achieve superhuman performance against a fixed opponent, their performance degrades substantially when confronted with a mixture of opponent strategies (Ellis et al., 2023). This lack of robustness significantly limits their applicability in realistic settings, where opponent strategies are often unknown, dynamic, and highly variable.

The problem is particularly pronounced in real-time strategy games such as StarCraft II. Within the SMAC benchmark, agents must coordinate to defeat scripted enemy teams, and the SMAC-Hard scenarios (Deng et al., 2024) introduce additional difficulty by incorporating diverse and adaptive opponent strategies. A key observation is that many MARL policies tend to overfit, exploiting the weaknesses of a single opponent rather than learning general and robust tactics. This highlights the need for mechanisms that enable agents to recognize the opponent's strategy and adapt their behavior accordingly.

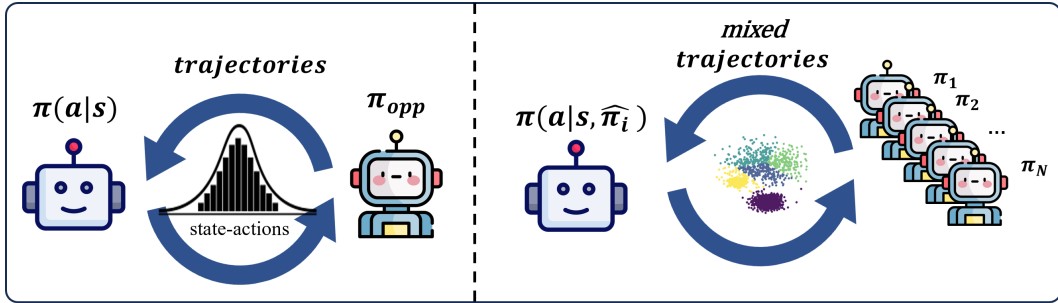

Figure 1: The left side depicts that a policy learns to interact with a static opponent policy, $\pi_{opp}$. The agent observes the state $s$ and takes an action $a$ from a distribution defined by $\pi(a|s)$, leading to an observed distribution of actions represented by the historical trajectory histogram. The right side illustrates training with "mixed trajectories." The policy learns from a diverse set of experiences generated by multiple opponent policies, $\pi_1$ to $\pi_N$. The mixed trajectories result in a more varied and complex set of observed states and actions, represented by the scattered points, which can lead to a more robust and generalized policy.

As illustrated in Figure 1, the agent's experience distribution depends heavily on the opponent's policy. Against a fixed opponent (left), the agent can infer the strategy by aggregating statistics over observed trajectories. However, in the mixed-strategy setting (right), trajectories from different opponents are interleaved, requiring the agent to infer the opponent's latent policy in real time to guide decision-making. Recent research directions, including opponent modeling (Yu et al., 2022b; Plaehn et al., 2023; Liang et al., 2023), sub-task assignment (Yang et al., 2022; You et al., 2025), and prototype-based contrastive learning (Li et al., 2020; Chen et al., 2021), have begun addressing this challenge. Nevertheless, these approaches remain limited, partly due to the lack of environments explicitly designed for training with mixed adversary strategies.

To address the opponent strategy identification issue and inspired by recent advances in representation learning and meta-reinforcement learning (Beck et al., 2023) where a latent context variable is learned to capture the core dynamics of a task, our method, Latent Opponent Strategy Identification (LOSI), consists of two core components: (1) an encoder that maps short observed opponent trajectories into low-dimensional embeddings, and (2) a conditioned MARL policy that adapts its decisions based on these embeddings. The encoder is trained using a contrastive learning objective (InfoNCE) (Gutmann & Hyvärinen, 2010), which does not require explicit strategy labels. The InfoNCE loss encourages embeddings from the same episode (thus the same opponent strategy) to cluster together, while pushing apart embeddings from different episodes. This allows the model to acquire discriminative representations of opponent strategies in an entirely unsupervised manner. The learned embeddings then condition both the MARL controller and the mixing network, enabling adaptive and robust decision-making against diverse opponents.

Therefore, our contributions are threefold: **1)** We propose a novel unsupervised approach for opponent strategy identification in MARL using contrastive learning, enabling discriminative representation learning without explicit labels. **2)** We demonstrate the effectiveness of LOSI by integrating it with standard MARL algorithms and evaluating it on challenging SMAC-Hard scenarios with mixed opponent strategies. **3)** We provide detailed analyses of the learned embedding space, showing that InfoNCE promotes clustering trajectories by strategy, even in the absence of ground-truth identifiers.

## 2 RELATED WORK

**Multi-agent Reinforcement Learning** In multi-agent value-based algorithms, the centralized value function, usually a joint Q-function, is decomposed into local utility functions. Many methods have been proposed to meet the Individual-Global-Maximum (IGM) (Son et al., 2019) assumption, which indicates the consistency between the local optimal actions and the optimal global joint action. VDN (Lowe et al., 2017) and QMIX (Rashid et al., 2018) introduce additivity and monotonicity to Q-functions. QTRAN (Son et al., 2019) transforms IGM into optimization constraints. QPLEX (Wang et al., 2020a) uses duplex dueling network architecture to guarantee IGM assumption. Instead

of focusing on value decomposition, multi-agent policy gradient algorithms provide a centralized value function to evaluate current joint policy and guide the update of each local utility network. Most policy-based MARL methods extend RL ideas, including MADDPG (Lowe et al., 2017), MATRPO (Kuba et al., 2021), MAPPO (Yu et al., 2022a). FOP (Zhang et al., 2021) algorithm factorizes optimal joint policy by maximum entropy and MACPF (Wang et al., 2023) is the latest algorithm that mixes critic values of each agent.

**Opponent Modeling and Policy Inference**   Opponent modeling is a core area of research for non-stationary multi-agent settings. Classic works by (He et al., 2016) introduced early neural network-based approaches to encode opponents' features and condition the agent's policy or Q-network on them. The Deep Policy Inference Q-Network (DPIQN) by (Hong et al., 2018) further formalized this idea by learning a feature representation of the opponent's policy from observations. More recently, works such as Model-Based Opponent Modeling (Yu et al., 2022b) have explored using learned environment models to anticipate opponent behavior and improve adaptation. These methods often require access to opponent observations, actions, or explicitly modeling the opponent's policies.

**Latent-Context and Task-Inference Methods**   Our work is also deeply connected to meta-reinforcement learning, which aims to learn a fast-adapting policy from a distribution of tasks. Probabilistic Embeddings for RL (PEARL) (Rakelly et al., 2019) is a prominent example, which learns a latent context variable to enable rapid adaptation to new tasks. The framework uses a variational inference approach to learn the latent variable from trajectory history. Similarly, Machine Theory of Mind (ToMnet) (Rabinowitz et al., 2018) uses a meta-learning approach to infer the beliefs and intentions of other agents. These works demonstrate the power of learning latent representations to disentangle task-specific information from the core policy. A related line of work is Learning Dynamic Subtask Assignment (LDSA) by (Yang et al., 2022) and DSRA (You et al., 2025), which addresses the problem of diverse agent behaviors in a cooperative team. LDSA and DSRA learns to assign agents to internal subtasks, which are learned from a subtask encoder based on the task identity, to encourage heterogeneity and improve collaboration.

**Contrastive and Prototype Contrastive Learning**   The use of contrastive learning for representation learning has seen a surge in popularity, particularly in computer vision and single-agent RL. Contrastive Learning in RL (CURL) (Laskin et al., 2020) demonstrated how InfoNCE-style objectives can be used to learn robust visual representations from raw pixel inputs, leading to more data-efficient learning. Prototypical Contrastive Learning (PCL) (Li et al., 2020) extended this by using prototypes to learn semantically meaningful clusters in the embedding space, which is conceptually similar to grouping different opponent strategies. These methods provide the methodological foundation for our encoder training.

## 3 BACKGROUNDS

A fully cooperative multi-agent task is described as a Dec-POMDP (Oliehoek et al., 2016) task which consists of a tuple $G = \langle S, A, P, r, Z, O, N, \gamma \rangle$ in which $s \in S$ is the true state of the environment and $N$ is the number of agents. At each time step, each agent $i \in N \equiv \{1, \ldots, n\}$ chooses an action $a_i \in A$ which forms the joint action $\mathbf{a} \in \mathbf{A} \equiv A^N$. The transition on the environment is according to the state transition function that $P(\cdot|s, \mathbf{a}) : S \times \mathbf{A} \times S \to [0, 1]$. The reward function, $r(s, \mathbf{a}) : S \times A \to \mathbb{R}$, is shared among all the agents, and $\gamma \in [0, 1)$ is the discount factor for future reward penalty.

Partially observable scenarios are considered in this paper that each agent draws individual observations $z \in Z$ of the environment according to the observation functions $O(s, i) : S \times N \to Z$. Meanwhile, the action-observation history, $\tau_i \in T \equiv (Z \times A)^*$, is preserved for each agent and conditions the stochastic policy $\pi_i(a_i|\tau_i) : T \times A \to [0, 1]$. The policy $\pi$ for each agent is determined by a joint action-value function: $Q^\pi(s^t, \mathbf{a}^t) = \mathbb{E}_{s^{t+1:\infty}, \mathbf{a}^{t+1:\infty}}[R^t|s^t, \mathbf{a}^t]$, in which the accumulated reward is considered as a discounted return and formulated as $R^t = \sum_{i=0}^{\infty} \gamma^i r^{t+i}$. After the rollout process, the whole trajectory from the initial transition to terminated transition $< (s_0, \mathbf{o}_0, \mathbf{a}_0, r_0), \ldots, (s_H, \mathbf{o}_H, \mathbf{a}_H, r_H) >$ are stored in the replay buffer.

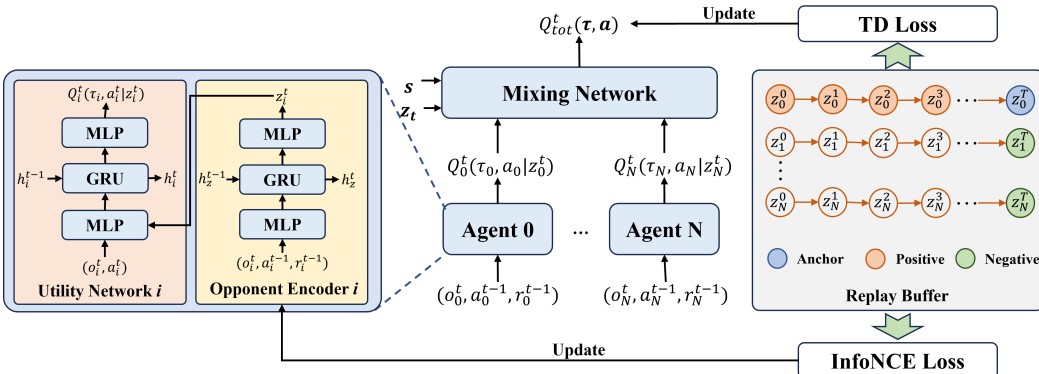

Figure 2: The utility networks and the mixing networks are based on original MARL algorithms. The Opponent encoders in our proposed method encode a window of past transitions of each timestep and are trained by contrastive loss during the training process. The opponent embeddings then serve as the conditions of the utility networks and the mixer network.

Deep q-learning algorithm aims to find the optimal joint action-value function $Q^*(s, \mathbf{a}; \theta) = r(s, \mathbf{a}) + \gamma \mathbb{E}_{s'} [\max_{\mathbf{a}'} Q^* (s', \mathbf{a}'; \theta)]$. Due to partial observability, $Q(\tau, \mathbf{a}; \theta)$ is used in place of $Q(s, \mathbf{a}; \theta)$ and parameters $\theta$ are learnt by minimizing the expected TD error. Centralized training and decentralized execution (CTDE) enables agents to acquire global states during the training and only individual observations during the testing execution. In multi-agent settings, VDN learns a joint action-value function $Q_{tot}(\tau, \mathbf{a})$ as the sum of individual value functions: $Q_{tot}^{\text{VDN}}(\tau, \mathbf{a}) = \sum_{i=1}^{n} Q_i(\tau_i, a_i)$. QMIX introduces a monotonic restriction $\forall i \in \mathcal{N}, \frac{\partial Q_{tot}^{\text{QMIX}}(\tau, \mathbf{a})}{\partial Q_i(\tau_i, a_i)} > 0$ to the mixing network to meet the IGM assumption. IGM asserts the consistency between joint and local greedy action selections in the joint action-value $Q_{tot}(\tau, \mathbf{a})$ and individual action-values $[Q_i(\tau_i, a_i)]_{i=1}^{n}$:

$$\arg\max_{\mathbf{a} \in \mathcal{A}} Q_{tot}(\tau, \mathbf{a}) = \begin{pmatrix} \arg\max_{a_1 \in \mathcal{A}} Q_1(\tau_1, a_1) \\ \vdots \\ \arg\max_{a_n \in \mathcal{A}} Q_n(\tau_n, a_n) \end{pmatrix}.$$

## 4 METHOD

We propose a framework, Latent Opponent Strategy Identification (LOSI), which augments value-based MARL algorithms with a contrastive identification module. LOSI is designed to stabilize latent representation learning and improve robustness against mixed opponent strategies by incorporating a prototype-guided InfoNCE objective. This section introduces the overall architecture of LOSI, details the latent opponent embedding mechanism, and explains the prototype-based contrastive learning procedure. The full pseudo-code is provided in the Appendix (Algorithm 1).

### 4.1 ARCHITECTURE

We focus on settings where opponent strategies are mixed and their true identities are not provided. To address this challenge, LOSI employs an opponent encoder that extracts latent strategy information from partial observations of the environment. Inspired by meta-reinforcement learning, the encoder processes agent-specific information: the current observation $o_i^t$, the previous action $a_i^{t-1}$, and the reward $r_i^{t-1}$, integrating temporal dependencies through Gated Recurrent Unit (GRU) cells. Within a fixed window size, the GRU accumulates trajectory information and outputs a latent embedding that summarizes the opponent's strategy from agent $i$'s perspective.

In value-based MARL algorithms, learning proceeds through temporal-difference (TD) updates of each agent's utility network. For methods such as QMIX, these updates are applied to the mixed value $Q_{tot}$. Each agent's utility network consists of MLP layers with recurrent GRU states $h_i^t$, which maintain trajectory history. Extending QMIX, LOSI augments the utility network with the predicted

opponent embedding $z_i^t$. At each timestep, the utility network of agent $i$ receives $o_i^t$, $a_i^t$, and $z_i^t$ as input, producing $Q_i(\tau_i, a_i | z_i^t)$. The individual utilities are then aggregated by the mixing network, which enforces monotonicity constraints through hyper-networks.

The mixing network is parameterized by a hyper-network that generates its weights and biases conditioned on both the global state $s$ and the averaged opponent embedding $\mathbf{z}_t$. For each layer in the mixer, the hyper-network outputs a set of weights and biases. These are combined with per-agent utilities $Q_i(\tau_i, a_i)$ and activated via ReLU functions to preserve monotonicity. The configuration of the mixing network follows the standard setup in PyMARL (Samvelyan et al., 2019).

Figure 2 illustrates the training process. Trajectory batches sampled from the replay buffer are used not only for TD learning but also for opponent strategy identification. The encoder is trained using a contrastive learning objective: embeddings from the same trajectory (positive samples) are encouraged to align, while embeddings from other trajectories in the batch (negative samples) are pushed apart. This contrastive signal ensures the learned embeddings capture discriminative features of opponent strategies.

## 4.2 LATENT EMBEDDINGS VIA CONTRASTIVE IDENTIFICATION

Our framework builds upon the value decomposition paradigm. In QMIX (Rashid et al., 2018), the joint action-value function is factorized into per-agent utilities using a monotonic mixing network:

$$Q_{tot}^t(\boldsymbol{\tau}, \mathbf{a}) = f_\theta\big(Q_1^t(o_1^t, a_1^t), \ldots, Q_n^t(o_n^t, a_n^t), s^t\big), \tag{1}$$

where $\boldsymbol{\tau}$ is the trajectory, $o_i^t$ and $a_i^t$, and $s^t$ denote agent observations, actions, global states at timestep $t$, and $f_\theta$ is constrained to be monotonic in its inputs. Each agent then selects decentralized actions greedily based on its individual utility. We extend this formulation by introducing an opponent encoder $E_\phi$, which maps sequences of observations, actions, and rewards to a latent embedding $z^t$:

$$z^t = E_\phi(\boldsymbol{\tau}^t). \tag{2}$$

The embedding $z_t$ conditions both the agent utility networks and the mixing network, providing a contextual signal that adapts the value function to the opponent's latent strategy. To capture temporal dynamics, the encoder is implemented as a recurrent GRU:

$$z_t = \text{GRU}_\phi(z^{t-1}, [o_i^t, a_i^{t-1}, r_i^{t-1}]), \tag{3}$$

During training, the encoder is optimized so that embeddings from trajectories generated by the same opponent strategy cluster together, while embeddings from different strategies remain well separated.

## 4.3 PROTOTYPE-BASED CONTRASTIVE LEARNING

To ensure that embeddings are discriminative and stable, we adopt a prototype-based InfoNCE objective. A memory bank maintains $K$ prototypes $p_k{}_{k=1}^K$, each representing a latent strategy cluster. For a given trajectory embedding $z$, the nearest prototype is identified as:

$$k^* = \arg\min_k \|z - p_k\|_2^2. \tag{4}$$

The contrastive loss is then defined as:

$$\mathcal{L}_{NCE} = -\log \frac{\exp\big(\text{sim}(z, p_{k^*})/\tau\big)}{\sum_{j=1}^K \exp\big(\text{sim}(z, p_j)/\tau\big)}, \tag{5}$$

where $\text{sim}(u, v) = \frac{u^\top v}{|u||v|}$ is cosine similarity and $\tau$ is a temperature parameter. Prototypes are updated online by exponential moving average:

$$p_{k^*} \leftarrow \alpha p_{k^*} + (1 - \alpha)z. \tag{6}$$

This objective encourages embeddings from the same opponent strategy to align with their nearest prototype, while preserving separation between different strategies. The prototype mechanism prevents collapse and stabilizes representation learning across training.

## 4.4 JOINT OPTIMIZATION

The final training objective combines temporal-difference (TD) learning with the prototype-based contrastive loss:

$$\mathcal{L}(\theta, \phi) = \mathcal{L}_{\text{TD}}(\theta, \phi) + \lambda(t) \, \mathcal{L}_{NCE}(\phi, \{p_k\}), \tag{7}$$

where the TD loss is defined as:

$$\mathcal{L}_{\text{TD}} = \mathbb{E}\Big[ \big( y_t - Q_{tot}(\boldsymbol{\tau}_t, \mathbf{a}_t; \theta, \phi) \big)^2 \Big], \tag{8}$$

with target values computed as:

$$y_t = r_t + \gamma \max_{\mathbf{a}'} Q_{tot}(\boldsymbol{\tau}_{t+1}, \mathbf{a}'; \theta^-). \tag{9}$$

The coefficient $\lambda(t)$ adaptively balances value learning and contrastive learning via an annealing schedule:

$$\lambda(t) = \lambda_{\max} \cdot \big(1 - e^{-\beta t}\big), \tag{10}$$

so that early training emphasizes stable value estimation, while later training places greater weight on embedding discrimination.

This joint optimization ensures that Q-learning faithfully optimizes the cooperative reward signal, while the embeddings remain clustered and strategy-aware. As a result, LOSI achieves stable convergence and strong generalization in mixed-strategy environments.

## 5 EXPERIMENT

In this section, we evaluate the performance of the proposed LOSI framework on fully cooperative SMAC-Hard micro-management challenges. Performance is measured by the mean winning rate in each scenario. We report results on six enhanced SMAC-Hard tasks and six newly designed, more complex scenarios. Additionally, we conduct ablation studies to assess the effectiveness of opponent strategy identification and the impact of ability usage.

### 5.1 EXPERIMENTS SETTINGS

**SMAC-Hard:** We first evaluate LOSI on six tasks from SMAC-Hard. Unlike standard SMAC tasks, SMAC-Hard introduces mixed opponent strategies, where opponents randomly select their behaviors from a strategy pool. Importantly, agents are not given access to the opponent strategy IDs. The difficulty level is set to 7 by default. Winning rates are averaged over five independent seeds and smoothed with a factor of 0.6 for better visualization across 10M time steps.

**SMAC-HARD with abilities:** To further validate LOSI, we design six additional tasks with increased complexity. These tasks incorporate fog of war, agent abilities, and larger agent populations, thereby enlarging both the joint action and observation spaces. Unlike the standard SMAC setting, ability usage is not restricted; units can freely use their abilities subject to cooldown constraints. As before, the difficulty level is set to 7, and winning rates are averaged over five seeds and smoothed with a factor of 0.6 within 10M time steps.

**Baselines:** We adapt our method to QMIX algorithms and compare our methods to the value-based QPLEX algorithm, popular policy-based algorithm MAPPO, and a dynamic sub-goal solution method DSRA. The QMIX, QPLEX in this paper are from the pymarl2 codebase Hu et al. (2021). MAPPO is provided by Yu et al. (2022a) and DSRA is from the github codebase You et al. (2025).

### 5.2 EXPERIMENT RESULTS

**SMAC-Hard** As illustrated in Figure 3, the proposed LOSI method consistently outperforms strong baseline algorithms, including QMIX, QPLEX, and MAPPO. Furthermore, LOSI also surpasses DSRA, a method specifically designed for handling multi-subgoal problems, highlighting its superior adaptability to diverse opponent behaviors. In the 10m_vs_11m task, the QPLEX algorithm experiences a sharp drop in winning rates, primarily due to the collapse of its attention matrix when

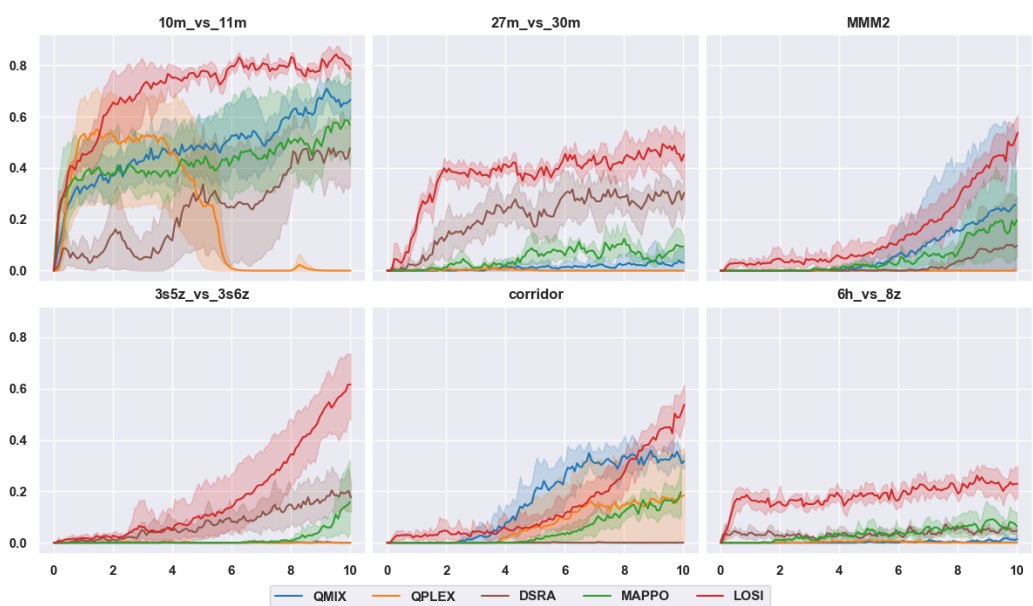

Figure 3: Comparison of our method against baselines on six SMAC-Hard scenarios with 5 mixed opponent strategies, including the original hard and super-hard tasks: 10m_vs_11m, 27m_vs_30m, MMM2, 3s5z_vs_3s6z, corridor, 6h_vs_8z. The solid line shows the average evaluation winning rate across 5 seeds and the shaded areas correspond to the 25-75% percentiles.

confronted with high performance variance across different runs. By contrast, LOSI maintains stable learning dynamics and achieves higher overall performance. Another particularly challenging case is the 6h_vs_8z task. Unlike most other scenarios, performance across all algorithms remains very low. Nevertheless, LOSI shows gradual improvement, eventually reaching a winning rate of around 23%, with a clear upward trend. This suggests that our method successfully identifies and defeats at least one of the opponent's strategies. However, due to the intrinsic difficulty of the task—arising from both the unit imbalance and the opponent's diverse tactical choices—longer training horizons may be required for LOSI to systematically adapt to and overcome the remaining strategies.

**SMAC-Hard with Abilities**   Figure 4 reports results on the six newly designed scenarios with ability usage. In these settings, opponent strategies involve stochastic activation of unit abilities, while agents can also employ abilities subject to their respective cooldown constraints, creating a significantly more dynamic and complex combat environment. Across all scenarios, LOSI demonstrates competitive or superior performance compared to the baselines. In 2vr_vs_3sc, 3rp_vs_5zl, and mmmt scenarios, LOSI achieves comparable results while still slightly outperforming other methods. In 7q_vs_2bc scenario, LOSI converges much faster, indicating its ability to rapidly recognize and adapt to opponent strategies. Notably, in the more demanding 3rp_vs_24zl and 7q_vs_2bc tasks, LOSI delivers substantial performance gains, significantly surpassing all baselines. These results highlight LOSI's ability to leverage the learned opponent embeddings to guide policy adaptation, even in high-dimensional action spaces involving ability management. Similar to the SMAC-Hard experiments, QPLEX occasionally suffers from attention collapse due to the large variance in outcomes, further underscoring LOSI's robustness and stability.

## 6   DISCUSSION

**Performance Enhancement**   As shown in Figure 3, removing the opponent identification module leads to clear performance degradation. In particular, the original value-based methods such as QMIX and QPLEX attempt to handle diverse opponent strategies using a single stationary policy. This design imposes significant challenges for accurate $Q$-value approximation. The centralized state-action values depend not only on the underlying environment state but also on the unobserved

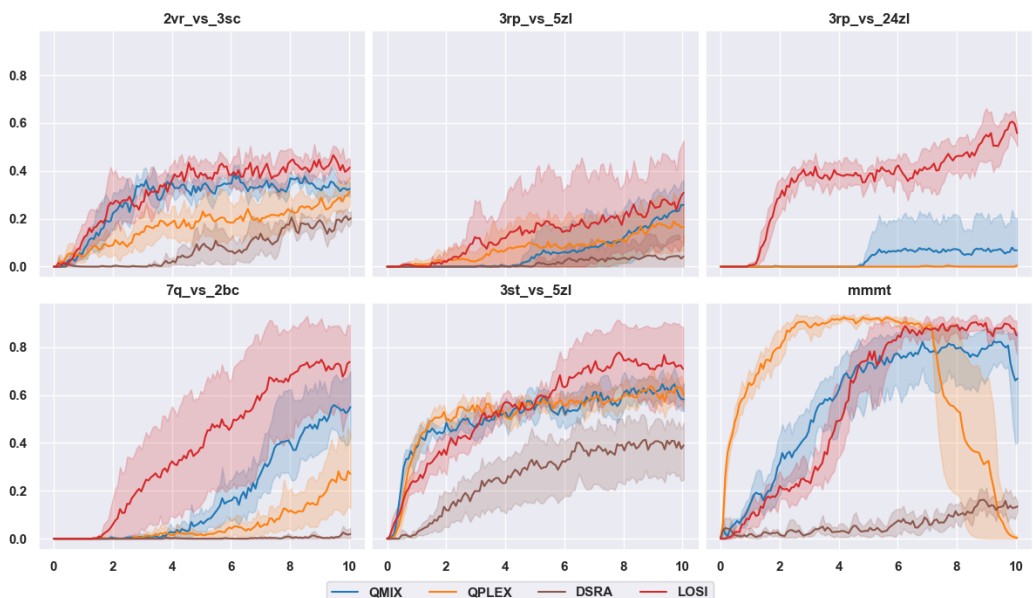

Figure 4: Comparison of our method against baselines on six newly-designed SMAC-Hard scenarios with 5 mixed opponent strategies. The codebase 'onpolicy' from MAPPO requires deeper modification of the starcraft.py which is not provided in the SMAC-Hard repository. Thus the MAPPO baseline is not provided here. The solid line shows the average evaluation winning rate across 5 seeds and the shaded areas correspond to the 25-75% percentiles.

opponent strategy. As a result, the $Q$ values associated with the same environmental state may vary considerably across trajectories when opponents adopt different strategies. This variance introduces instability into both the centralized mixing network and the individual policy networks. A similar phenomenon is observed in the actor–critic paradigm, where MAPPO suffers from unstable value estimates in the critic network, which in turn negatively affects the quality of the learned actor policies.

The DSRA algorithm, despite its ability to model multiple subgoals, fails to extend this capacity to effective opponent strategy recognition. DSRA employs a variational autoencoder (VAE) to generate subgoal embeddings and reconstruct observations, after which the embeddings are combined with an ability matrix and passed through a Gumbel-Softmax operation for policy selection. However, DSRA faces two major limitations in this setting. First, the number of subgoals must be specified as a fixed hyperparameter, while in practice the number of opponent strategies is unknown and potentially variable. Second, the embeddings learned by DSRA often shift rapidly within a single trajectory, which can undermine the efficiency and stability of subgoal identification. Together, these factors limit the applicability of DSRA when tackling dynamic opponent behaviors in mixed-strategy environments.

**Agent Abilities** Figure 4 highlights the experimental results for scenarios in which both agents and opponents can exploit specialized abilities. Unlike the standard SMAC-Hard tasks, these new tasks allow agents to make decisions about whether and when to use abilities, such as the Blink ability for Stalker units, which enables instantaneous relocation. These abilities greatly expand the effective action space and introduce additional layers of strategic diversity. Both agents and opponents can employ abilities at varying times and on different targets, which significantly increases the complexity of policy learning. Importantly, this enlarged action space provides more fine-grained signals for meta-learning, thereby facilitating more precise opponent strategy identification.

In these settings, the timing and targeting of abilities become as critical as traditional movement and attack decisions. Thus, the training process requires not only accurate $Q$-value estimates for standard actions but also highly reliable estimates for ability-related decisions, which often serve as decisive actions in combat. Unstable $Q$ values for these critical actions can lead to large

fluctuations in performance, making it particularly challenging for baseline algorithms to converge. Our LOSI framework, by contrast, demonstrates superior robustness in such scenarios, as its opponent identification module enables agents to adaptively condition their policies on inferred opponent strategies. This capability explains its strong performance in the ability-enabled environments, where relying on a single stationary policy is insufficient to achieve competitive results.

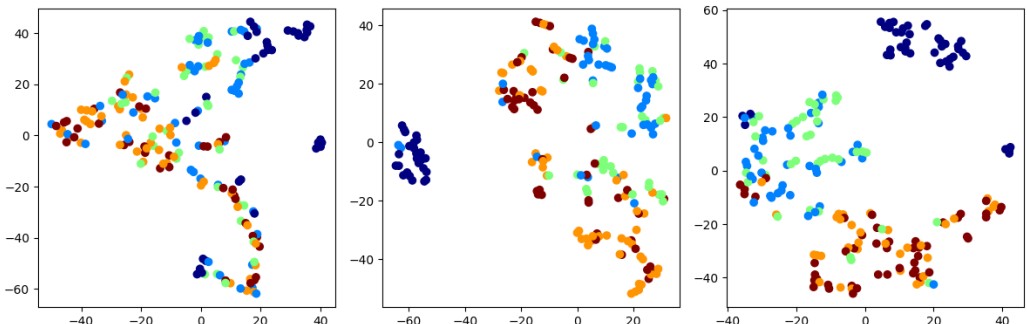

Figure 5: The t-SNE visualization results of the opponent policy embedding at the first 200, 10000 to 10020, and 20000 to 20200 training iterations.

**Opponent Strategy Identification**  To further validate the effectiveness of our framework, we visualize the learned embeddings of opponent strategies using t-SNE (Maaten & Hinton, 2008). As illustrated in Figure 5, the embeddings are initially intermingled, reflecting the difficulty of distinguishing opponent behaviors at the beginning of training. After approximately 10k iterations, the model successfully separates one opponent strategy, and by the final 200 episodes, three distinct clusters emerge. Notably, while the SMAC-Hard environment contains five ground-truth opponent strategies, our LOSI identifies three robust clusters. This suggests that some strategies, such as "attack the nearest enemy" and "attack the weakest enemy," may be too similar in practice to yield separable embeddings.

Importantly, the clustering observed in t-SNE does not rely on explicit access to opponent strategy IDs, which are hidden from the agent during training. Instead, the embeddings reflect an implicit disentanglement of strategic patterns that are aligned with the convergence dynamics of the utility networks. Despite the partial mismatch between the number of ground-truth strategies and the number of clusters identified, the visualization demonstrates that LOSI captures meaningful strategic distinctions. This implicit recognition process contributes directly to performance improvements, as agents are better equipped to adjust their behaviors based on inferred opponent tactics.

## 7 CONCLUSION AND FUTURE WORK

In this paper, we introduced Latent Opponent Strategy Identification (LOSI), a novel framework that enables agents in multi-agent reinforcement learning (MARL) to dynamically adapt their policies to diverse opponent strategies. Departing from traditional approaches, LOSI learns discriminative representations of opponent behaviors in a fully unsupervised manner by training an encoder with a contrastive learning objective (InfoNCE). This design effectively clusters sub-trajectories from the same episode while separating those from different episodes, producing latent embeddings that serve as powerful conditioning signals for MARL policies and mixing networks. Through extensive experiments on the challenging SMAC-Hard benchmarks, we showed that LOSI substantially outperforms strong baselines, delivering improved robustness and generalization across mixed and evolving opponent strategies. Notably, LOSI achieves this without requiring explicit strategy labels, which represents an important step toward developing adaptive and strategy-aware agents for complex real-world applications. Looking forward, we plan to extend LOSI to a wider range of MARL algorithms, including both value-decomposition and actor–critic architectures with value-based critics, and to integrate it with online opponent modeling or meta-learning techniques for faster adaptation.

## 8 CHECKLIST

### 8.1 DECLARATION OF LLM USAGE

During the paper writing, LLMs are used solely for polishing the writing, such as correcting spelling and grammar errors, and for no further purpose.

### 8.2 ETHICS

There are no ethical concerns currently because the codebase, the environment, and the data are open-sourced and are cited in the paper.

### 8.3 REPRODUCIBILITY

The testbed is publicly accessible from GitHub, and StarCraft II is provided by Storm platform. The codes are also provided in the supplementary materials.

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

## A  PSEUDOCODE

The pseudocode is provided here according to the descriptions in the method section.

---

**Algorithm 1** QMIX with Prototype-based Contrastive Identification

---

**Require:** Agent networks $Q_\theta^i$, mixing network $f_\theta$, encoder $E_\phi$, prototype memory $\{p_k\}_{k=1}^K$, replay buffer $\mathcal{D}$
  **for** each training episode **do**
    Reset environment, initialize hidden states $h_0^i$, encoder state $z_0$
    **for** each timestep $t = 1, \ldots, T$ **do**
      Each agent $i$ selects $a_t^i \sim \pi^i(o_t^i, h_t^i, z_t)$
      Execute joint action $\mathbf{a}_t$, observe $r_t$, next obs $\mathbf{o}_{t+1}$
      Update hidden states $h_{t+1}^i$ and encoder embedding $z_{t+1} = E_\phi(\boldsymbol{\tau}_{t+1})$
      Store transition $(\boldsymbol{\tau}_t, \mathbf{a}_t, r_t, \boldsymbol{\tau}_{t+1}, z_t)$ in $\mathcal{D}$
    **end for**
    Sample batch $\mathcal{B}$ from $\mathcal{D}$
    **for** each episode in $\mathcal{B}$ **do**
      Compute encoder embedding $z = E_\phi(\boldsymbol{\tau})$
      Find nearest prototype $p_{k^*} = \arg\min_k \|z - p_k\|^2$
      Compute InfoNCE loss:

$$\mathcal{L}_{\text{NCE}} = -\log \frac{\exp(\text{sim}(z, p_{k^*})/\tau)}{\sum_j \exp(\text{sim}(z, p_j)/\tau)}$$

      Update prototype $p_{k^*} \leftarrow \alpha p_{k^*} + (1 - \alpha)z$
    **end for**
    Compute TD targets $y_t = r_t + \gamma \max_{\mathbf{a}'} Q_{tot}(\boldsymbol{\tau}_{t+1}, \mathbf{a}')$
    Compute TD loss $\mathcal{L}_{\text{TD}} = (y_t - Q_{tot}(\boldsymbol{\tau}_t, \mathbf{a}_t))^2$
    Form joint objective $\mathcal{L} = \mathcal{L}_{\text{TD}} + \lambda(t)\,\mathcal{L}_{\text{NCE}}$
    Update $\theta, \phi$ by gradient descent on $\mathcal{L}$
  **end for**

---

## B  SMAC-HARD

The experiment testbed in this paper is SMAC-Hard scenario, which is an enhanced version of original SMAC tasks in the aspect of mixed opponent strategies. The SMAC-Hard environment is a new environment, which might not be known to public. Therefore, we briefly summarize the content of SMAC-Hard.

### B.1  SUMMARY

The paper 'SMAC-Hard: Enabling Mixed Opponent Strategy Script and Self-play on SMAC' introduces an extended and more challenging benchmark for Multi-Agent Reinforcement Learning (MARL) to address the issue of algorithms overfitting to the static, limited opponent policies in the original StarCraft Multi-Agent Challenge (SMAC). The core problem identified is that existing SMAC adversaries lack sufficient diversity, causing MARL agents to learn strategies that exploit unintended vulnerabilities rather than developing true robustness and generalizability.

To overcome these limitations, the authors propose SMAC-HARD, a novel framework that significantly increases the complexity of the opponent strategies. The benchmark's main enhancements include support for customizable opponent strategies, randomization of adversarial policies, and unit abilities support. These features expose the agents to a broad and dynamic spectrum of opponent behaviors during training, forcing them to learn more stable and universally applicable policies.

The experimental evaluations conducted using SMAC-HARD demonstrated that even widely used and state-of-the-art MARL algorithms struggle significantly against the newly introduced edited and mixed-strategy opponents. The paper concludes by positioning SMAC-HARD as a critical next step in MARL research, one that is necessary for properly benchmarking and advancing the development of the next generation of algorithms capable of operating reliably in unpredictable, open-ended multi-agent systems.

## B.2 EXAMPLE SCRIPTS

The SMAC-Hard provides different LLM-genreated and manually edited opponent scripts. In this subsection, according to the authors' official github repository, we analyze and demonstrate some example opponent script pieces to show the diverse opponent strategy.

### B.2.1 BASE SCRIPT

```
...
def script(self, agents, enemies, agent_ability, visible_matrix, iteration):

    actions = []
    agents = [agent for _, agent in agents.items() if agent.health != 0]
    enemies = [enemy for _, enemy in enemies.items() if enemy.health != 0]
    if not agents or not enemies:
        return []

    if self.map_name in ['3st_vs_5zl', '3rp_vs_5zl', '2c_vs_64zg']:
        for a in agents:
            actions.append(attack(a, (16, 16), visible_matrix))
...
```

### B.2.2 ATTACK NEAREST ENEMY

```
...
    weakest_agent = min(agents, key=lambda e: e.health / e.health_max)

    for agent in agents:

        if agent.unit_type == UnitTypeId.MEDIVAC.value:
            actions.append(move(agent,(weakest_agent.pos.x+2, weakest_agent.pos.y)))
        else:
            target_tag = self.target_dict.get(agent.tag, None)
            target = find_by_tag(enemies, target_tag)
            if target == None or target.health == 0:

                nearest_enemy = min(enemies, key=lambda e: distance_to(agent, e))
                self.target_dict[agent.tag] = nearest_enemy.tag
                target_tag = nearest_enemy.tag

            actions.append(attack(agent, find_by_tag(enemies, target_tag), visible_matrix))
...
```

### B.2.3 ATTACK WEAKEST ENEMY

```
...
    weakest_enemy = min(enemies, key=lambda e: (e.health + e.shield) / (e.health_max + e.
     shield_max))
        weakest_agent = min(agents, key=lambda e: e.health / e.health_max)

        for agent in agents:
            if agent.unit_type == UnitTypeId.MEDIVAC.value:
                actions.append(move(agent,(weakest_agent.pos.x, weakest_agent.pos.y)))
            else:

                target_tag = self.target_dict.get(agent.tag, None)
                target = find_by_tag(enemies, target_tag)
                if target == None or target.health == 0:

                    self.target_dict[agent.tag] = weakest_enemy.tag
                    target = weakest_enemy

                actions.append(attack(agent, target, visible_matrix))
...
```

### B.2.4   3RP_VS_5ZL SCRIPT 1

```
...
    if self.init:
        self.up_cliff = [agents[0].tag, agents[1].tag]
        self.down_cliff = [agents[2].tag, agents[3].tag]
        self.free = agents[4].tag
        self.targets = {}
        self.init=False

    if len(agents) >= 4:
        # Group 1 on the cliff and Group 2 off the cliff, the free agent always chases enemies
        for agent in agents:
            nearest_target = min(enemies, key=lambda e: distance_to(e, agent))
            if agent.tag in self.up_cliff:
                if nearest_target.pos.y > self.cliff_y:
                    actions.append(attack(agent, nearest_target, visible_matrix))
                else:
                    actions.append(move(agent, (16, 16)))
            elif agent.tag in self.down_cliff:
                if nearest_target.pos.y < self.cliff_y:
                    actions.append(attack(agent, nearest_target, visible_matrix))
                else:
                    actions.append(move(agent, (16, 9)))
            elif agent.tag ==self.free:
                actions.append(attack(agent, nearest_target, visible_matrix))
    else:
        for iter, agent in enumerate(agents):
            target = find_by_tag(enemies, self.targets.get(agent.tag, None))
            if target == None or target.health == 0:
                target = min(enemies, key=lambda e: distance_to(e, agent))
                self.targets[agent.tag] = target.tag
            actions.append(attack(agent, target, visible_matrix))

    return actions
...
```

### B.2.5   3RP_VS_5ZL SCRIPT 2

```
...
    if self.init:
        for agent in agents:
            self.status[agent.tag] = 'Attack'
        self.init = False

    # Assign targets to agents in groups
    for a_id, agent in enumerate(agents):

        if agent.health / agent.health_max < 0.2 and agent.shield/agent.shield_max < 0.2:
            self.status[agent.tag] = 'Retreat'
        elif agent.shield/agent.shield_max >0.8:
            self.status[agent.tag] = 'Attack'

        status = self.status[agent.tag]

        target = find_by_tag(enemies, self.groups.get(agent.tag, None))
        if target == None or target.health == 0:
            self.groups[agent.tag] = enemies[a_id % len(enemies)].tag
            target = enemies[a_id % len(enemies)]

        if status == 'Attack':
            actions.append(attack(agent, target, visible_matrix))
        elif status == 'Retreat':
            actions.append(move(agent, toward(agent, target, -2)))

    return action
...
```

### B.2.6 SCRIPT WITH ABILITIES

```
...
    for agent in agents:

        a_ability = [ab.abilities for ab in agent_ability if ab.unit_tag==agent.tag][0]
        avail_ability = [a.ability_id for a in a_ability]

        if agent.unit_type == 140:
            # Uprooted Spore Crawler
            target = min(enemies, key=lambda e: distance_to(e, agent))

            if distance_to(agent, self.center) > 5:
                actions.append(move(agent, self.center))
            else:
                if distance_to(agent, target) >= 8 and distance_to(agent, self.center) < 4:
                    actions.append(move(agent, target))
                else:
                    if 1731 in avail_ability:
                        actions.append(apply_ability(agent, 3680, (agent.pos.x, agent.pos.y)))
        elif agent.unit_type ==99:
            # rooted Spore Crawler
            target = min(enemies, key=lambda e: distance_to(e, agent))
            if distance_to(target, agent) >= 8 or distance_to(agent, self.center) >5:
                if 1727 in avail_ability:
                    actions.append(apply_ability(agent, 3681, None))
            else:
                actions.append(attack(agent, target, visible_matrix))

    return actions
...
```

## C HYPER-PARAMETERS

In this paper, we conduct experiments on the five algorithms, including QMIX, QPLEX, DSRA, MAPPO, and LOSI. The performance is influenced by hyper-parameters used in the experiments. In this section, we list the hyper-parameters we use in during the training. Because of the large amount of parameters, we only show the important different parameters compared to the default settings. The QMIX and QPLEX algorithms implementation are from the pymarl2 codebase which follows the pymarl2 default parameters. The MAPPO is from the on-policy codebase, in which the authors provide specific parameters for each scenario in SMAC, so we use these hyper-parameters in the scripts.

Table 1: Hyper-parameters in experiments

| hyper-parameter | QMIX | QPLEX | DSRA | LOSI |
|---|---|---|---|---|
| runner | parallel | parallel | episode | parallel |
| batch_size_run | 8 | 8 | 1 | 8 |
| batch_size | 128 | 128 | 32 | 128 |
| t_max | 10050000 | 10050000 | 10050000 | 10050000 |
| epsilon_anneal_time | 100000 | 100000 | 50000 | 100000 |
| td_lambda | 0.6 | 0.6 | 0.0 | 0.6 |
| optim | Adam | Adam | RMSProp | Adam |
| double_q | False | True | True | True |
| adv_hypernet_layers | - | 3 | - | - |
| adv_hypernet_embed | - | 64 | - | - |
| num_kernel | - | 10 | - | - |
| is_minus_one | - | True | - | - |
| weighted_head | - | True | - | - |
| is_adv_attention | - | True | - | - |
| is_stop_gradient | - | True | - | - |
| n_subtasks | - | - | 4 | - |
| agent_subtask_embed_dim | - | - | 64 | - |
| sft_way | - | - | gumbel_softmax | - |
| lambda_subtask_prob | - | - | 0.001 | - |
| lambda_subtask_repr | - | - | 0.001 | - |
| subtask_policy_use_hypernet | - | - | True | - |
| use_tanh | - | - | True | - |
| subtask_repr_layers | - | - | 2 | - |
| random_sele | - | - | False | - |
| z_dim | - | - | - | 16 |
| ctx_window | - | - | - | 10 |
| ctx_hid | - | - | - | 64 |
| ctx_reward | - | - | - | True |
| nce_negatives | - | - | - | 64 |
| nce_queue_size | - | - | - | 5000 |
| nce_momentum | - | - | - | 0.9 |
| nce_temperature | - | - | - | 0.07 |

