# OpenReview forum: "LOSI: Improving Multi-agent Reinforcement Learning via Latent Opponent Strategy Identification"
_ICLR.cc/2026/Conference — ICLR 2026 Conference Withdrawn Submission_

### Official Review · Reviewer_nbH8 · 2025-10-27

**Soundness:** 2
**Presentation:** 1
**Contribution:** 1
**Rating:** 2
**Confidence:** 4

**Summary:**

This paper introduces Latent Opponent Strategy Identification (LOSI), a framework designed to improve the generalization of cooperative Multi-Agent Reinforcement Learning (MARL) agents against a mixture of opponent strategies. LOSI employs a trajectory encoder trained with a prototype-based contrastive learning objective (InfoNCE) to map observed opponent trajectories into a latent embedding space. This learned embedding is then used as a conditioning signal for both the individual agent policies and the central mixing network. The paper evaluates LOSI on the SMAC-Hard benchmark and show that it outperforms several strong MARL baselines.

**Strengths:**

Clear presentation and structure. The empirical section is easy to follow, and the figures show learning curves with performance margin and visual analysis of the embedding space.

**Weaknesses:**

1. Limited novelty. Learning a latent context for opponent behavior and conditioning the policy is a well-established idea in opponent modeling. The contribution reads as an incremental combination rather than a new principle.
2. Insufficient coverage of opponent modeling literature and baselines. Since opponent modeling is the core contribution, the paper should cite and compare against more direct baselines. Current comparisons (QMIX, QPLEX, MAPPO, DSRA) do not isolate the benefit of opponent modeling versus strong, established opponent-modeling approaches.
3. Narrow evaluation. All experiments are within the SMAC family (SMAC-Hard and ability variants). There is no evidence of generalization to other domains.
4. Computational overhead is not discussed. The added encoder introduces training and inference costs. There is no report of wall-clock time, parameter counts, or throughput relative to the baselines. This matters for practical adoption.

**Questions:**

1. What is the training and inference overhead introduced by LOSI?
2. How sensitive is performance to the number of prototypes, window length, embedding dimension, and negatives?
3. Does LOSI generalize to other environments beyond SMAC?

---

### Official Review · Reviewer_aXXJ · 2025-10-31

**Soundness:** 3
**Presentation:** 2
**Contribution:** 2
**Rating:** 4
**Confidence:** 4

**Summary:**

The paper studies multi-agent reinforcement learning with diverse opponent strategies. The paper proposes learning latent embeddings of opponents' strategies based on observed trajectories, ensuring discriminative representations of different strategies by using prototype-based contrastive learning. These latent embedding are then used as an addition input to enhance the q-learning part. Experiments on SMAC-hard show that the proposed method outperforms baselines.

**Strengths:**

1. The paper tackles a challenging MARL problem in which opponent strategies are diverse. The proposed idea of incorporating discriminative latent embeddings for different opponent strategies into q learning is promising.
2. Experiment results on SMAC-hard show the proposed ideas enhance the policy learning outcomes substantially compared to strong baselines such as QMIX, QPLEX, MAPPO, etc.

**Weaknesses:**

1. The discussion on related work on opponent modeling is limited. The paper only mentioned 3 papers on opponent modeling in the related work. However, there is a long line of research works focusing on opponent modeling in MARL. Some of them have similar ideas of opponent latent embedding. A couple of examples include [*] and [**]. It is important for the paper to position itself in comparison with such existing works.

* Papoudakis, G. and Albrecht, S.V., 2020. Variational autoencoders for opponent modeling in multi-agent systems. arXiv preprint arXiv:2001.10829.
** Xie, Annie, Dylan Losey, Ryan Tolsma, Chelsea Finn, and Dorsa Sadigh. "Learning latent representations to influence multi-agent interaction." In Conference on robot learning, pp. 575-588. PMLR, 2021.

2. The experiments are limited to only the SMAC environment. It is unclear how the proposed method's performance can be generalized to other MARL environments.

**Questions:**

1. How can we determine the number of prototypes given the number of opponent strategies are not known in advance?

2. What does the difficulty level of SMAC represent and how that will influence the performance of the proposed method?

---

### Official Review · Reviewer_sriT · 2025-11-01

**Soundness:** 2
**Presentation:** 3
**Contribution:** 3
**Rating:** 6
**Confidence:** 4

**Summary:**

The paper proposes a novel unsupervised approach for opponent strategy identification in MARL using contrastive learning, enabling discriminative representation learning without explicit labels. It demonstrates the effectiveness of LOSI by integrating it with standard MARL algorithms and evaluating it on challenging SMAC-Hard scenarios with mixed opponent strategies. The paper provides detailed analyses of the learned embedding space, showing that InfoNCE promotes clustering trajectories by strategy, even in the absence of ground-truth identifiers.

**Strengths:**

- The paper deals with an important problem of the MARL community; that is opponent modelling and generalization in cooperative-competitive settings
- The paper proposes a novel and very simple methodology to train opponent modelling embeddings based on contrastive learning
- The proposed method seems to significantly improve baselines in SMAC benchmarks.
- The t-SNE visualization results are very interesting, highlighting that the proposed method indeed verifies the authors' claims.

**Weaknesses:**

- The only technical contribution of this paper is to train the modelling embeddings with the NCE loss. Although it seems that this modification significantly improves established methods, it would be very beneficial to conduct experiments on more benchmarks, such as Google Football, and also compare with other state-of-the-art opponent modelling methods (see for example [1-2]).
- Since the method relies on more clever opponent modelling and policy enhancement through opponent modelling, important related work is missing, see [1-3]. The general idea of policy enhancement through opponent modelling is not new, so a more extended discussion would be very beneficial.
- To further validate the effectiveness of the proposed framework, it would be nice to test it on top of other value-based methods, such as QPLEX.
- The selection of K needs to be better justified, as it is not clear how one should select K and how K should be interpreted.

**References**:

[1] Kontogiannis, Andreas, et al. "Enhancing Cooperative Multi-Agent Reinforcement Learning with State Modelling and Adversarial Exploration." Forty-second International Conference on Machine Learning. 2025.

[2] Papoudakis, Georgios, Filippos Christianos, and Stefano Albrecht. "Agent modelling under partial observability for deep reinforcement learning." Advances in Neural Information Processing Systems 34 (2021): 19210-19222.

[3] Sun, J., Chen, S., Zhang, C., Ma, Y., and Zhang, J. Decisionmaking with speculative opponent models. IEEE Transactions on Neural Networks and Learning Systems, 2024.

**Questions:**

- How should one select K? I believe that selecting K requires knowing a priori the pool of the different mixed strategies that the players may select. Have the authors tried a more adaptive way to select K, without knowing a priori such information?
- How are the prototypes initialized? Have the authors evaluated different initialized schemes?

---

### Official Review · Reviewer_gjwM · 2025-11-01

**Soundness:** 1
**Presentation:** 2
**Contribution:** 2
**Rating:** 2
**Confidence:** 4

**Summary:**

This paper presents Latent Opponent Strategy Identification (LOSI) to address existing limitations in contemporary MARL methods, which easily overfit to policy against a fixed opponent strategy. In particular, LOSI employs a trajectory encoder trained with a contrastive learning objective to discriminate the opponents’ behaviors.

**Strengths:**

This paper studies the relatively under-explored subject, opponent modeling for conditional policy learning in MARL.

**Weaknesses:**

See the questions and comments.

**Questions:**

**Questions**

Q1. In Figure 3, the unit of timestep should be properly presented. Is it mil (1e6)?

Q2. How to control opponents’ strategies? The author mentioned five opponent strategies. What are they? How are they sampled?

Q3. The recent work TRAMA [1], which identifies distinct trajectories for both allies and enemies, should be discussed and compared. The overall framework is quite similar to the proposed method.

[1] Na, H., Lee, K., Lee, S. and Moon, I.C., 2025, March. Trajectory-Class-Aware Multi-Agent Reinforcement Learning. In The Thirteenth International Conference on Learning Representations.

Q4. For InfoNCE loss computation, the author mentioned “the InfoNCE loss encourages embeddings from the same episode to cluster together while pushing apart embeddings from different episodes.” However, couldn’t it be possible that opponents may have the same strategy in different episodes?

Q5. In LOSI, is PCL directly adopted to learn trajectory embedding? Or there was a modification?

Q6. How does $z_t$ in Eq. (3) encode the opponent’s strategy? It is just the embedding of the observation and reward trajectory of the $i$-th agent. By the way, $z_t$ in Eq. (3) should be $z_i^t$, or it should include other agents’ information.

Q7. How to define $K$? Is the proposed model robust to $K$? A corresponding sensitivity study or at least some discussions should be presented.

Q8. Do we use supervision about which opponent strategy is used to generate a given trajectory during training? Then, we choose the nearest $p_k$ during execution?

Q9. How to construct $z_t$ from $z_i^t$s?

**Comments**

C1. Missing space in the caption in Figure 1, “defined by$\pi(a|s)$.

C2. Please check expression for $[p_k]_{k=1}^K$ in Section 4.3.

---

### Official Review · Reviewer_vcYb · 2025-11-03

**Soundness:** 2
**Presentation:** 3
**Contribution:** 1
**Rating:** 2
**Confidence:** 4

**Summary:**

This paper proposes Latent Opponent Strategy Identification (LOSI), a framework to improve generalization in collaborative Multi-Agent Reinforcement Learning (MARL) against diverse and unseen opponent strategies. The core problem addressed is that many MARL algorithms overfit to specific opponent behaviors, leading to poor performance when strategies change. LOSI uses a trajectory encoder trained with an unsupervised, prototype-based contrastive learning (InfoNCE) objective. This encoder learns to map opponent behaviors into latent embeddings. These embeddings are then used to condition both the individual agent policies and the mixing network, enabling adaptive decision-making. The authors evaluate LOSI on challenging SMAC-Hard scenarios with mixed opponent strategies, demonstrating improved performance over baselines like QMIX, QPLEX, and DSRA.

**Strengths:**

The paper addresses an important and well-known problem in MARL: generalization against diverse opponent strategies. The experimental results, on their face, are a strength. The method shows better performance and win rates than several MARL baselines (QMIX, QPLEX, MAPPO, DSRA) across a suite of SMAC-Hard scenarios, including newly designed tasks with abilities.

**Weaknesses:**

Despite the positive results, the paper suffers from significant weaknesses in its methodology, novelty, and experimental evaluation.

1.  **Methodological Unclearity and Non-Stationarity:** The core premise is flawed. The method is called "Latent Opponent Strategy Identification," but the architecture diagram (Figure 2) and text state that the "Opponent Encoder $i$" takes agent $i$'s *own* local trajectory $(o_i^t, a_i^{t-1}, r_i^{t-1})$ as input. It's unclear how this encodes the *opponent's* strategy, especially in partially observable settings. If "opponent" is meant to include teammates, the method fails to address the non-stationarity of *those* agents. The contrastive objective contrasts *episodes*, assuming a fixed strategy within an episode. But in MARL, teammates' policies are also updating. This means the representations learned from the replay buffer are "lagging" and based on outdated policies. The paper does not address this fundamental conflict.

2.  **Limited Novelty:** The contribution appears to be a straightforward application of contrastive unsupervised learning (like CURL [1]) to a standard MARL algorithm (QMIX). Using an encoder to learn representations from trajectories via InfoNCE and then conditioning a policy on those representations might be a good pattern. The paper does not sufficiently motivate why this is a non-trivial extension.

3.  **Missing Critical Baselines:** This is the most significant weakness.
    * **Missing Opponent Modeling Baselines:** The paper cites several key opponent modeling works in its related work section (e.g., He et al., 2016; Hong et al., 2018; Yu et al., 2022b) but provides zero empirical comparison. We are left with no way to know if this complex contrastive framework is any better than simpler, established opponent modeling techniques.
    * **Missing Contrastive MARL Baselines:** The paper completely ignores a large and growing body of recent work that explicitly combines MARL with contrastive learning (e.g., Hu et al., 2024 [2]; Lo et al., 2024 [3]; Song et al., 2023 [4]; Xu et al., 2023 [5]; Liu et al., 2023 [6]). These papers are highly relevant and must be discussed and compared against to properly situate LOSI's contribution.

4.  **Lack of Ablation Studies:** The method introduces several new components and hyperparameters, but their impact is never analyzed. The model uses a prototype-based InfoNCE loss, an annealing schedule $\lambda(t)$, and conditions both the utility networks and the mixer. There are no ablations to test the effect of the contrastive loss itself (vs. no loss), the importance of the prototypes (vs. standard InfoNCE), or the sensitivity to the number of prototypes $K$.

[1] Laskin M, Srinivas A, Abbeel P. Curl: Contrastive unsupervised representations for reinforcement learning. ICML, 2020.

[2] Hu Z, Zhang Z, Li H, et al. Attention-Guided Contrastive Role Representations for Multi-agent Reinforcement Learning. ICLR, 2024.

[3] Lo Y L, Sengupta B, Foerster J N, et al. Learning Multi-Agent Communication with Contrastive Learning. ICLR. 2024.

[4] Song H, Feng M, Zhou W, et al. MA2CL: masked attentive contrastive learning for multi-agent reinforcement learning. IJCAI, 2023.

[5] Xu Z, Zhang B, Li D, et al. Consensus learning for cooperative multi-agent reinforcement learning. AAAI, 2023.

[6] Liu S, Zhou Y, Song J, et al. Contrastive identity-aware learning for multi-agent value decomposition. AAAI, 2023.

**Questions:**

1.  Could you clarify the input to the "Opponent Encoder $i$"? Figure 2 and Section 4.1 suggest it takes agent $i$'s own local history. How does this reliably encode the *opponent's* strategy?
2.  The core contrastive loss operates on trajectories from the replay buffer, sampled from different episodes. How does the framework account for the non-stationarity of *teammate* policies? It seems the learned representations would be "lagging" as they are based on old, outdated policies from the buffer.
3.  Why were there no empirical comparisons to the opponent modeling baselines (like Yu et al., 2022b) that you cite in the related work?
4.  Can you discuss the novelty of LOSI with respect to other recent MARL papers that also use contrastive learning for representation (e.g., [2, 3, 4, 5, 6] in my review)?
5.  The t-SNE visualization shows roughly 3 clusters emerging, but you state the environment has 5 ground-truth strategies. How was the number of prototypes $K$ chosen for the experiments, and how sensitive is the model's performance to this hyperparameter? A proper abaltion seems necessary.

---

### Official Review · Reviewer_ZG7d · 2025-11-08

**Soundness:** 2
**Presentation:** 3
**Contribution:** 2
**Rating:** 2
**Confidence:** 2

**Summary:**

This paper addresses environments such as SMAC-Hard, where opponent agents' strategies may change, and proposes a method called LOSI. LOSI identifies the strategies of opponents to help the agent cope with the non-stationarity of opponent strategies. Additionally, it employs contrastive learning to better distinguish between opponent strategies.

**Strengths:**

The paper presents a relatively clear explanation of the motivation and the specific method. The authors also include some hyperparameters of the different algorithms in the experiment section in the appendix.

**Weaknesses:**

Overall, the validation of the method's effectiveness is somewhat insufficient.
The following issues exist:
1. Could the number of prototypes set be critical? If a large number is set but only a few are actually used, would this affect training?
2. How are the prototypes initialized? Could this have a significant impact on subsequent training, such as the final number of actual clusters?
3. Are the baseline algorithms comparable to LOSI in terms of parameter count? Could the improvement in LOSI's performance be due to its increased network parameters, leading to better generalization? If the baseline algorithms use their original parameter sizes in such more complex environments, the comparison might be unfair.
4. The baseline algorithms do not include other meta-learning methods.
5. According to Fig. 3 and Fig. 4, some algorithms have not yet converged in certain environments. Should training be extended for these more challenging tasks?
6. There is a lack of ablation studies. How would the algorithm perform if contrastive learning is removed and only the encoder is retained?

**Questions:**

See Weaknesses.

---

### Note · Authors · 2025-12-04

**Comment:**

We sincerely appreciate the time and effort that the reviewers have dedicated to evaluating our manuscript. After carefully considering the reviewers’ comments and suggestions, we recognize that addressing some of the raised questions would require additional experiments and analyses that cannot be completed within the current rebuttal period. In light of this, we have decided to withdraw our manuscript at this time. We are grateful for the constructive feedback provided, which will be invaluable in guiding our revisions and future work. We hope to resubmit an improved version of the manuscript in the upcoming venues.

**Withdrawal Confirmation:**

I have read and agree with the venue's withdrawal policy on behalf of myself and my co-authors.